# Adipose HuR protects against diet-induced obesity and insulin resistance

Jingyuan Li[1], Li Gong[2], Shaozhuang Liu[3], Yujie Zhang[1], Chunmei Zhang[1], Mi Tian[1], Huixia Lu[1], Peili Bu[1], Jianmin Yang[1], Changhan Ouyang[4], Xiuxin Jiang[3], Jiliang Wu[4], Yun Zhang[1], Qing Min[4], Cheng Zhang[1] & Wencheng Zhang[1,4]

Human antigen R (HuR) is a member of the Hu family of RNA-binding proteins and is involved in many physiological processes. Obesity, as a worldwide healthcare problem, has attracted more and more attention. To investigate the role of adipose HuR, we generate adipose-specific *HuR* knockout (HuR[AKO]) mice. As compared with control mice, HuR[AKO] mice show obesity when induced with a high-fat diet, along with insulin resistance, glucose intolerance, hypercholesterolemia and increased inflammation in adipose tissue. The obesity of HuR[AKO] mice is attributed to adipocyte hypertrophy in white adipose tissue due to decreased expression of adipose triglyceride lipase (ATGL). HuR positively regulates ATGL expression by promoting the mRNA stability and translation of *ATGL*. Consistently, the expression of HuR in adipose tissue is reduced in obese humans. This study suggests that adipose HuR may be a critical regulator of ATGL expression and lipolysis and thereby controls obesity and metabolic syndrome.

[1] The Key Laboratory of Cardiovascular Remodeling and Function Research, Chinese Ministry of Education, Chinese National Health Commission and Chinese Academy of Medical Sciences, The State and Shandong Province Joint Key Laboratory of Translational Cardiovascular Medicine, Department of Cardiology, Qilu Hospital of Shandong University, Jinan 250012, China. [2] Department of Traditional Chinese Integrated Western Medicine, Qilu Hospital of Shandong University, Jinan 250012, China. [3] Department of General Surgery, Qilu Hospital of Shandong University, Jinan 250012, China. [4] School of Pharmacy, Hubei University of Science and Technology, Xianning 437100 Hubei, China. Correspondence and requests for materials should be addressed to Q. M. (email: baimin0628@163.com) or to C.Z. (email: zhangc@sdu.edu.cn) or to W.Z. (email: zhangwencheng@sdu.edu.cn)

O besity is a chronic metabolic disease that is becoming increasingly common. It favors many metabolic complications including type 2 diabetes, hypertension and cardiovascular diseases[1,2]. The critical characteristic of obesity is excessive triglycerides (TG) storage in adipose tissue, which is achieved by adipocyte hyperplasia (increased number) or hypertrophy (increased size) or even both. Adipocyte hypertrophy is believed to occur before adipocyte hyperplasia and to be the main mechanism of fat mass expansion[3,4].

Lipolysis is a process of triglycerides sequential hydrolysis to produce glycerol and free fatty acids (FFAs) in cell lipid droplets, which maintains the TG-FFA constant flow together with lipogenesis[5–7]. Considering that lipolysis is critical to supply glycerol and fatty acids as energy substrates to tissues, aberrant adipose lipolysis would be associated with lipodystrophy, hyperlipidemia or obesity[8,9].

Adipose triglyceride lipase (ATGL), also called desnutrin or patatin-like phospholipase domain containing 2, is considered the main enzyme responsible for the lipolytic process. It catalyzes the first step of lipolysis and converts TG to diacylglycerol and FFAs[10–13]. ATGL is expressed predominantly in white adipose tissue (WAT) and brown adipose tissue (BAT) and is mainly located in the lipid droplets of adipocytes[6]. Global ATGL-deficient mice displayed a severe basal and stimulated lipolytic defect and systemic TG overload in WAT and nonadipose tissue, which highlighted the crucial role of ATGL in lipolysis[14,15]. Accumulating evidence has demonstrated that several factors participate in the regulation of ATGL gene expression. Peroxisome proliferator activated receptor (PPAR) agonists type α and γ, AMP-activated protein kinase and glucocorticoids could elevate the mRNA level of ATGL, whereas insulin, mammalian target of rapamycin (mTOR) complex 1 activation and food intake could reduce its mRNA level[15–19].

Human antigen R (HuR), a ubiquitously expressed member of the Hu family of RNA-binding proteins, contains highly conserved recognition motifs and selectively binds adenylate-uridylate-rich element (AREs) usually found in the 3′-untranslated region (UTR) of its targets, conferring stability of ARE-containing mRNAs[20–23]. In vitro studies have shown that HuR regulates the expression of genes involved in key cellular processes such as inflammation, stress response and apoptosis[24–27]. Global HuR-knockout mice exhibited embryonic lethality due to defects in placental development[28]. Intestinal deletion of HuR gene caused a decreased tumor burden in models of intestinal tumorigenesis and inflammatory colon carcinogenesis[29]. B lineage-specific deletion of HuR led to impaired survival of B cells in bone marrow and antibody production of all isotypes, which affected humoral immunity[30]. However, the specific role of HuR in adipose tissue has not been clearly elucidated.

In this study, we generate adipose-specific HuR-knockout (HuRAKO) mice by using a Cre-loxP strategy and investigate the role of HuR in adipose tissue. Adipose-specific HuR ablation predisposes mice to high-fat diet (HFD)-induced obesity and insulin resistance.

## Results

**Adipose-specific HuR ablation sensitizes mice to obesity.** To determine the function of HuR in adipose tissue, we first evaluated whether its expression in adipose tissue could be changed by nutritional challenge. We detected HuR expression in WAT, including epididymal (epiWAT, visceral) and inguinal (ingWAT, subcutaneous) fat pads as well as BAT. The protein and mRNA levels of HuR were significantly decreased in WAT and BAT from the leptin mutant (ob/ob) and HFD-fed mice, the models of obesity and type 2 diabetes, as compared with their controls

(Fig. 1a, b and Supplementary Fig. 1a, b). Thus, the expression of HuR appeared to be negatively associated with obesity in mice. The dynamics of HuR expression prompted us to explore whether this RNA-binding protein could regulate energy metabolism in adipose tissue.

To assess the role of HuR specifically in adipose tissue, we generated adipose-specific HuR-knockout (HuRAKO) mice by crossbreeding HuRflox/flox mice with adipoQ-derived Cre transgenic mice (Fig. 1c). The protein and mRNA levels of HuR were significantly decreased in adipose tissues of HuRAKO mice (Fig. 1d, e), which was further confirmed by immunohistochemistry assay (Supplementary Fig. 1e). As expected, the expression of HuR was not changed in liver, muscle or other tissues of HuRAKO mice (Fig. 1e). Consistently, HuR expression was decreased by approximately 90% in mature adipocytes of adipose tissue from HuRAKO mice (Fig. 1f) but not in the stromal vascular fraction (SVF) (Fig. 1g), the source of preadipocytes and macrophages.

HuRAKO mice did not exhibit overt abnormalities. The 8-week-old HuRAKO mice and their control littermates were then fed a normal chow diet or HFD for 16 weeks. When challenged with HFD, HuRAKO mice gained more weight and had higher fat mass than their controls (Fig. 2a–c). At 24 weeks of age, HuRAKO mice had significantly greater epiWAT and ingWAT fat mass relative to control mice ($2.31 \pm 0.10$ vs. $1.66 \pm 0.08$ g, $P < 0.001$; $1.78 \pm 0.42$ vs. $0.81 \pm 0.10$ g, $P < 0.05$; Significance was determined by Student's $t$ test analysis), whereas BAT mass was slightly but not significantly increased in HuRAKO mice (Fig. 2d). Furthermore, HuRAKO mice showed higher serum levels of total cholesterol, triglycerides and low-density lipoprotein (LDL) and lower level of high-density lipoprotein (HDL) than controls (Fig. 2e). Together, these data indicate that adipose-specific ablation of HuR predisposes to HFD-induced obesity and lipid metabolism disorders.

**HuR ablation results in adipocyte hypertrophy.** An increase in adipose tissue mass can be attributed to an increase in adipocyte size or number due to abnormal differentiation, or both. To reveal the mechanism of increased adiposity in HuRAKO mice, we measured adipocyte size in adipose tissue of HFD-fed control and HuRAKO mice. H&E staining indicated that adipocytes were larger in both epiWAT and ingWAT of HuRAKO than control mice (Fig. 3a). The increased adipocyte size in HuRAKO adipose tissue was further supported by cell size quantification (Fig. 3b). Besides, HuR overexpression or knockout did not affect the adipose differentiation (Supplementary Fig. 2a,b), thereby suggesting that increased fat mass in HuRAKO mice was caused by adipocyte hypertrophy.

We also evaluated the effect of HuR ablation in adipose tissue on basic metabolic activity. Under the HFD condition, HuRAKO mice showed significantly reduced oxygen consumption and heat production, increased respiratory exchange rate (RER) as compared with controls (Fig. 3c−e and Supplementary Fig. 2c−e). Control and HuRAKO mice did not differ in food intake and physical activity (Supplementary Fig. 2f,g). The reduced resting metabolic rate might account for the obesity seen in HuRAKO mice.

To explore the specific mechanism of HuR ablation causing adipocyte hypertrophy, we used gene array assay of control and HuR-deficient adipose tissue. We found 103 genes with >1.5-fold upregulation and 113 genes with >1.5-fold downregulation with HuR deletion (Fig. 3f and Supplementary Data 1). We consulted the mRNA 3′-UTR region of 113 downregulated genes and found that there were 75 genes with those 3′-UTR region containing AREs as the predicted targets of HuR (Supplementary Table 2). We further measured the mRNA level of lipid metabolism-

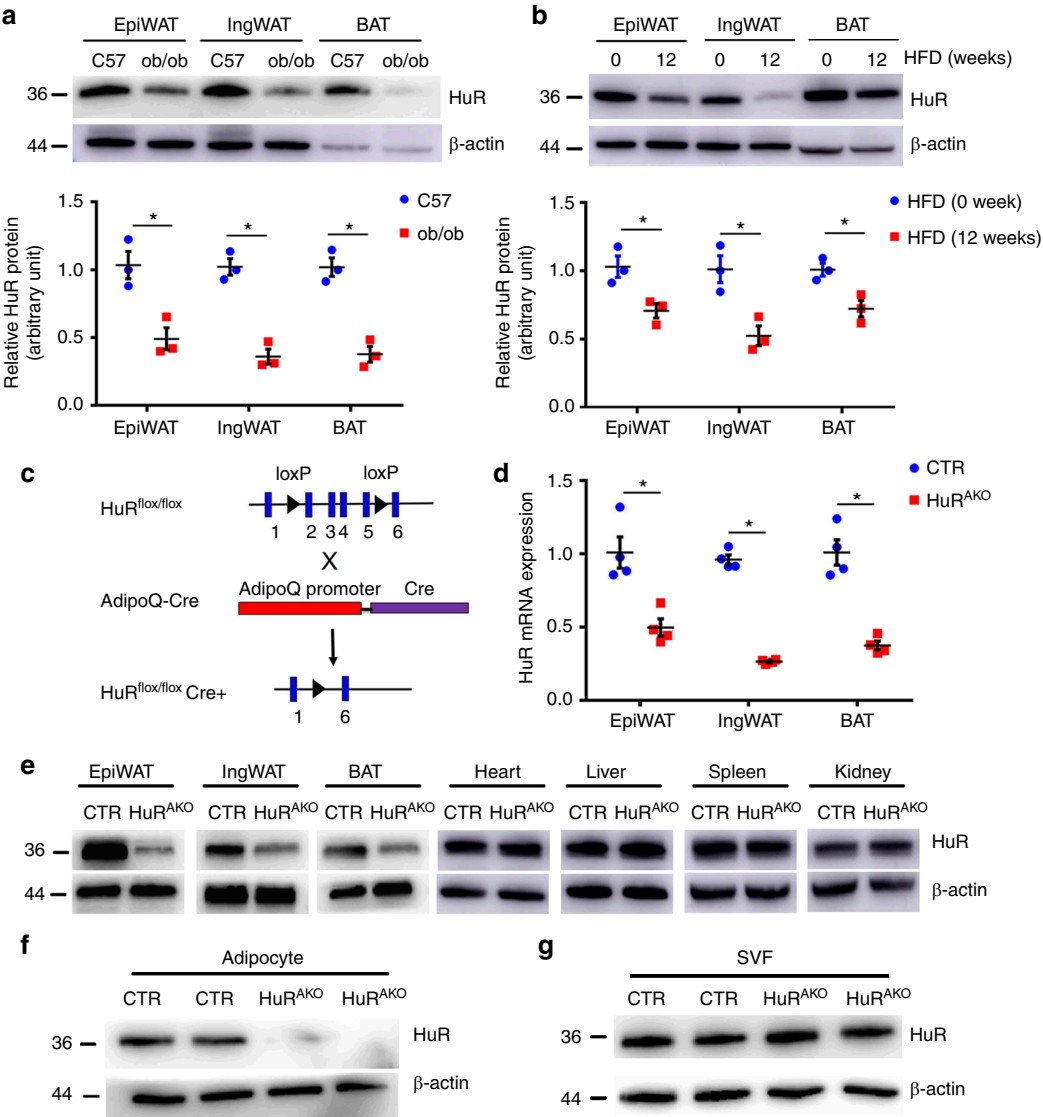

**Fig. 1** Generation of adipose-specific *HuR*-knockout mice. **a** Western blot analysis of HuR protein expression in adipose tissue (epiWAT, ingWAT and BAT) of 20-week-old C57BL/6J and ob/ob mice and quantification ($n = 3$), *comparison of ob/ob vs. C57. **b** Eight-week-old male C57BL/6J mice were fed with an HFD for an additional 12 weeks; western blot analysis of HuR and β-actin in adipose tissue and quantification ($n = 3$), *comparison of HFD (12w) vs. HFD (0w). **c** Schematic diagram of transgenic mice used to generate HuR[AKO] mice. **d** qPCR analysis of *HuR* mRNA expression in adipose tissue from control and HuR[AKO] mice ($n = 4$), *comparison of HuR[AKO] vs. control. **e–g** Western blot analysis of HuR protein level in tissues (**e**) and adipocytes (**f**) and stromal vascular fraction (SVF) (**g**) from control and HuR[AKO] mice. Data are represented as mean ± SEM. Significance was determined by Student's *t* test analysis, *$P < 0.05$. Source data are provided as a Source Data file

associated markers in adipose tissue by qPCR. The expression of adipogenic transcription factors, including CCAAT/enhancer-binding protein α (*C/EBPα*), and *PPARγ* were not changed in HuR[AKO] mice, nor were the expression of adipocyte differentiation markers such as adipocyte fatty acid-binding protein (*Fabp4*) and lipoprotein lipase (*Lpl*) and fatty acid synthesis genes such as acetyl-CoA carboxylase (*ACC*). However, lipolysis-associated genes including *ATGL* and Perilipin (*PLIN*) were significantly inhibited in HuR[AKO] mice as compared with controls (Fig. 3g). The protein expression of ATGL and PLIN was also decreased with *HuR* ablation in adipose tissues (Fig. 3h). These data suggest that adipocyte hypertrophy in HFD-fed HuR[AKO] mice may be due to inhibited lipolysis and greater triglycerides accumulation rather than abnormal adipocyte differentiation.

**Adipose-specific *HuR* ablation suppresses adipocyte lipolysis.**
To further determine whether lipolytic activity was impaired in

HuR[AKO] mice, the epiWAT from HFD-fed control and HuR[AKO] mice was isolated and treated with isoproterenol (ISO, β adrenergic receptor agonist), which simulates lipolysis by activating the protein kinase A (PKA) pathway. ISO-stimulated glycerol and FFA release were decreased in HuR[AKO] explants vs. control explants. Both basal and stimulated FFA release rates were reduced with HuR ablation (Fig. 4a). We further measured lipolysis stimulated by CL316243 (β3 adrenergic receptor agonist) in vivo. Serum FFA as well as glycerol levels were significantly lower in HuR[AKO] than control mice 15 min after CL316243 stimulation (Fig. 4b).

In light of the adversity of cell populations within adipose explants, we isolated epididymal SVFs from HuR[AKO] and control mice and stimulated their differentiation into adipocytes followed by lipolytic assay. As expected, ISO-stimulated FFA and glycerol release amount and rate were decreased in differentiated adipocytes from HuR[AKO] mice (Fig. 4c). In addition, we

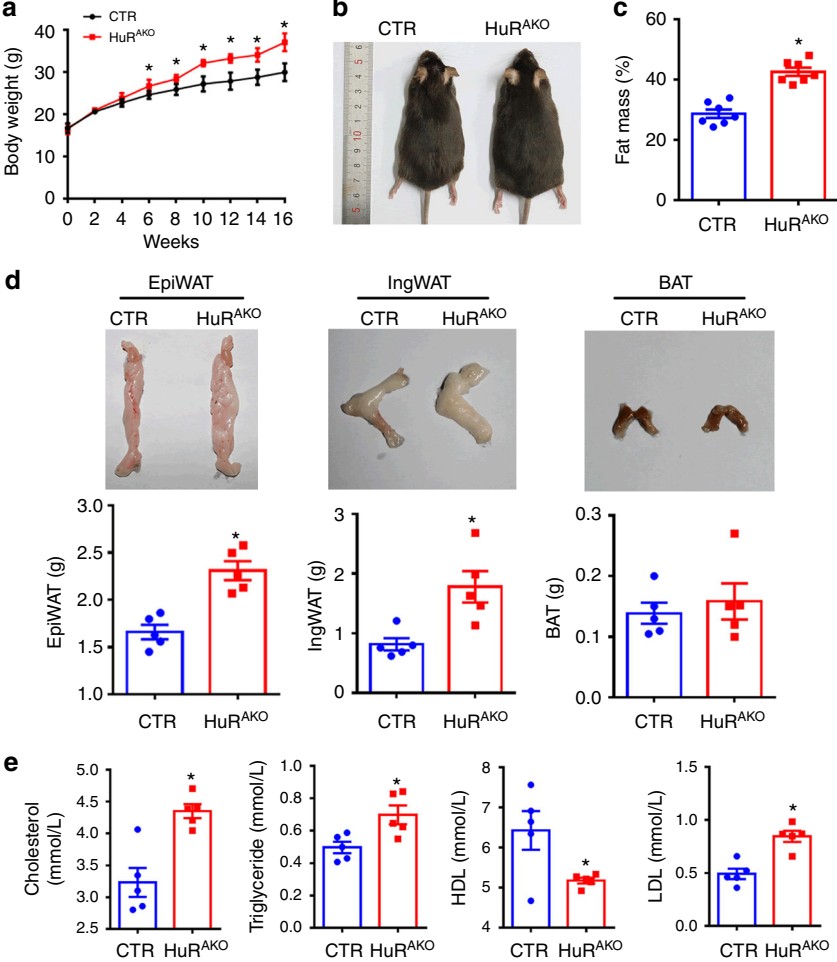

**Fig. 2** Adipose-specific *HuR* ablation sensitizes mice to obesity. **a** Body weight of control and HuR^AKO mice fed an HFD (*n* = 10), *comparison of HuR^AKO vs. control. **b** Representative photographs of control and HuR^AKO mice fed an HFD. **c** Fat mass analysis of control and HuR^AKO mice fed an HFD (*n* = 7), *comparison of HuR^AKO vs. control. **d** Representative photographs of fat pads (epiWAT, ingWAT and BAT) (top) and quantification (bottom) (*n* = 5), *comparison of HuR^AKO vs. control. **e** Serum lipid profiles (cholesterol, triglycerides, HDL and LDL) in control and HuR^AKO mice fed an HFD (*n* = 5), *comparison of HuR^AKO vs. control. Data are represented as mean ± SEM. Significance was determined by Student's *t* test analysis, **P* < 0.05. Source data are provided as a Source Data file

evaluated the effect of HuR overexpression on lipolysis. 3T3-L1 adipocytes were infected with an adenovirus expressing HuR or green fluorescent protein (GFP) and stimulated with ISO. HuR overexpression significantly exacerbated ISO-stimulated lipolysis (Fig. 4d). These results demonstrate that *HuR* ablation in adipose tissue suppressed lipolysis, which resulted in adipocyte hypertrophy and adiposity in HuR^AKO mice.

**HuR regulates *ATGL* mRNA stability.** In light of the suppressed lipolysis and obesity phenotype in HuR^AKO mice, we examined the expression of lipolysis markers in ob/ob and HFD-fed C57/BL mice. The protein and mRNA levels of *ATGL* were decreased in these two obesity models, which were consistent with those of HuR (Fig. 5a, b and Supplementary Fig. 1c, d). To further determine the relation between HuR and ATGL, SVFs were isolated from control and HuR^AKO mice and then stimulated to differentiate into adipocytes. ATGL expression was decreased with *HuR* ablation (Fig. 5c). In addition, *HuR* overexpression increased ATGL protein level in differentiated 3T3-L1 adipocytes (Fig. 5d).

To better understand the regulation of ATGL expression by HuR, we examined the 3′-UTR of *ATGL* to assess whether *ATGL*

may be a target of post-transcriptional modulation. We identified one conserved ARE in the 3′-UTR of mice *ATGL* mRNA, which suggests that *ATGL* may be a target of HuR. 3T3-L1 adipocytes were treated with the HuR-ARE interaction inhibitor CMLD-2, which can disrupt the interaction between HuR and its target mRNA. ATGL protein level was inhibited by CMLD-2 (Fig. 5e). We assessed the ability of HuR to bind to *ATGL* mRNA. RNA immunoprecipitation (IP) assay with an anti-HuR antibody or IgG revealed that HuR could bind to *ATGL* mRNA, which was inhibited by CMLD-2 (Fig. 5f). The regulation of *ATGL* mRNA by HuR was further confirmed by half-life assay. Differentiated SVFs from control or HuR^AKO mice were treated with actinomycin D, a transcriptional inhibitor. We found that the half-life of *ATGL* mRNA was shorter in HuR^AKO than control adipocytes (Fig. 5g). Moreover, consistent with the downregulation of HuR under high-fat diet, the *ATGL* mRNA stability in the epididymal adipose tissue from HFD-fed mice was decreased compared with that from normal chow diet (NCD)-fed mice (Supplementary Fig. 3a). However, HuR overexpression did not affect the half-life of *PLIN* (Supplementary Fig. 3b), which indicated that *PLIN* is not an HuR target. Above all, these results demonstrated that HuR directly targets *ATGL* mRNA and increases its stability and protein level.

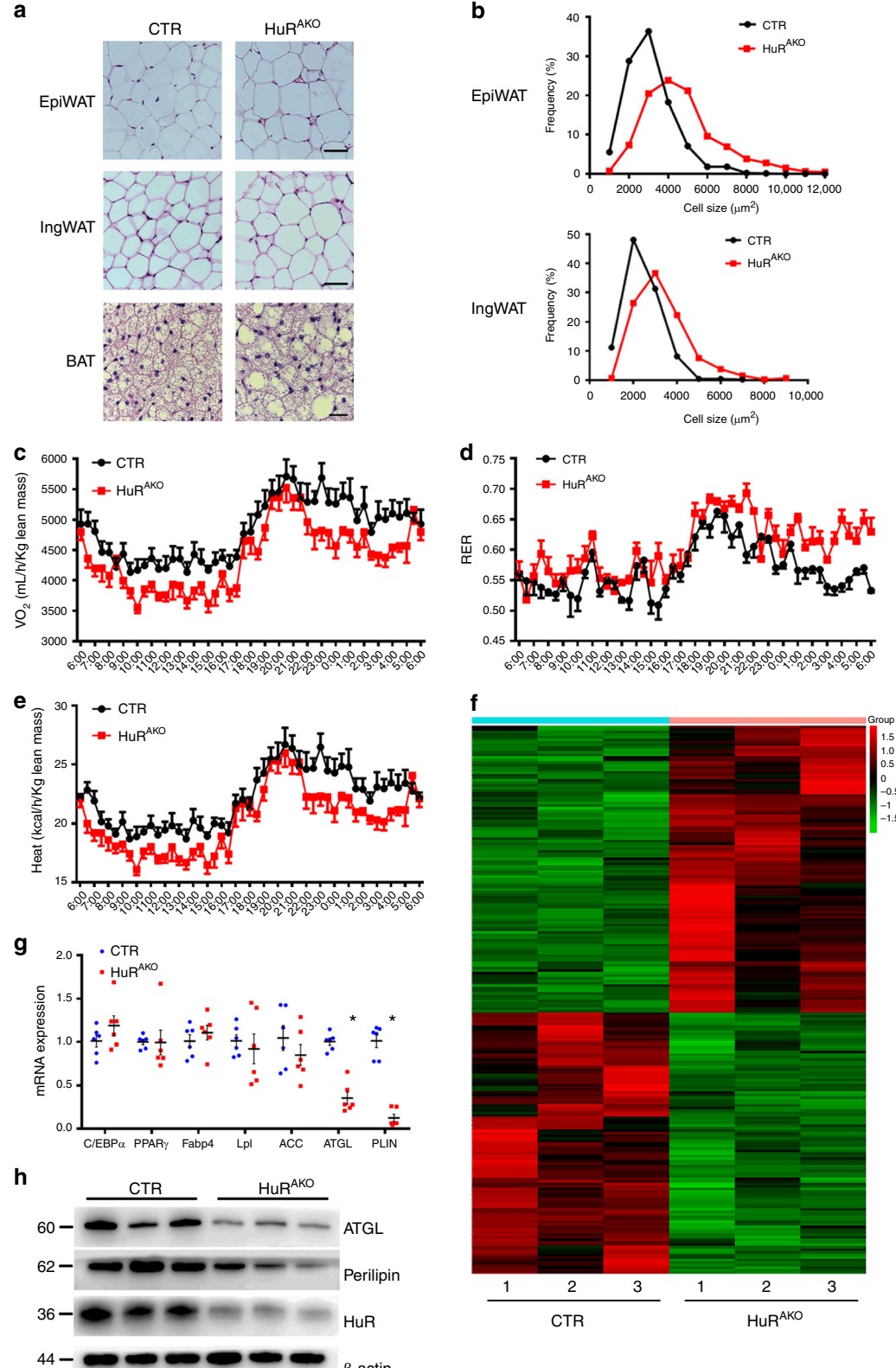

**Fig. 3** *HuR* ablation results in adipocyte hypertrophy. **a** Representative H&E images of epiWAT, ingWAT and BAT in HFD-fed control and HuR[AKO] mice. Scale bar 50 μm for WAT and 20 μm for BAT. **b** Quantification of adipocyte diameter. Total 300–350 cells per group were measured (*n* = 5). **c–e** Metabolism studies of control and HuR[AKO] mice fed an HFD: oxygen consumption (**c**), respiratory exchange ratio (RER) (**d**) and heat production (**e**) (*n* = 4). **f** Heatmap represents genes with >1.5-fold upregulation or >1.5-fold downregulation in *HuR*-deficient adipose tissue compared with control. **g** qPCR analysis of mRNA levels of genes related to adipogenesis, adipocyte differentiation and lipolysis in epididymal adipose tissue from control and HuR[AKO] mice fed an HFD (*n* = 6), *comparison of HuR[AKO] vs. control. **h** Protein expression of ATGL and PLIN with *HuR* ablation in adipose tissue (*n* = 3). Data are represented as mean ± SEM. Significance was determined by Student's *t* test analysis, *$P < 0.05$. Source data are provided as a Source Data file

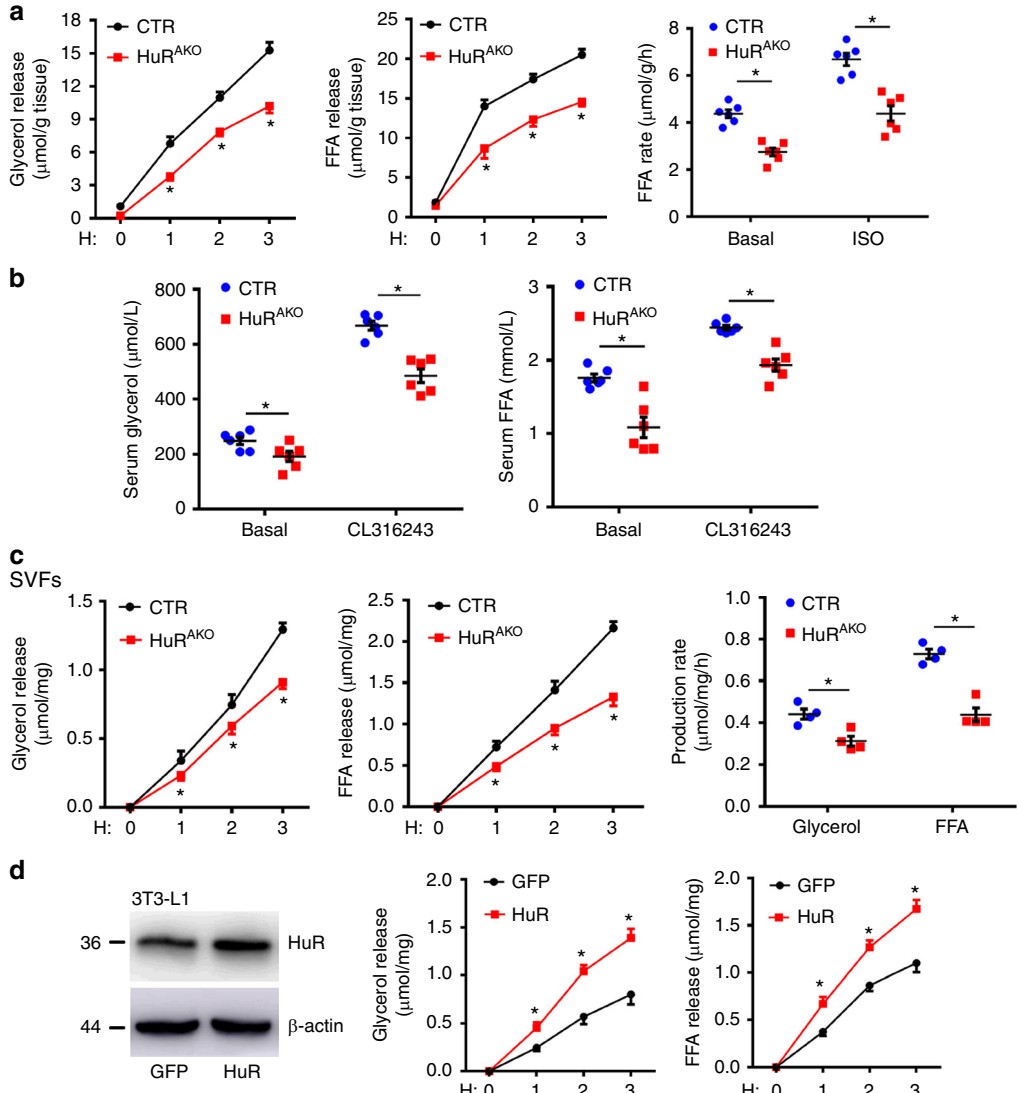

**Fig. 4** Adipose-specific *HuR* ablation suppresses adipocyte lipolysis. **a** EpiWAT explants were prepared from HFD-fed control and HuR^AKO mice, and stimulated with 1 μM ISO. FFA and glycerol release were measured (normalized to total protein levels). FFA release rates were calculated (*n* = 6), *comparison of HuR^AKO vs. control. **b** HFD-fed control and HuR^AKO mice were injected with CL316243 (1 mg kg⁻¹ body weight). Serum FFA and glycerol levels were measured (*n* = 6), *comparison of HuR^AKO vs. control. **c** SVFs were isolated from control and HuR^AKO mice, differentiated to adipocytes, and stimulated with 10 μM ISO. Glycerol and FFA release were calculated (normalized to total protein levels) (*n* = 4), *comparison of HuR^AKO vs. control. **d** Differentiated 3T3-L1 adipocytes were infected with adenovirus expressing GFP or HuR for 48 h. Cell extracts were immunoblotted with antibodies against HuR and β-actin. Adipocytes were stimulated with 10 μM ISO, and FFA and glycerol release were calculated (normalized to total protein levels) (*n* = 4), *comparison of HuR vs. GFP. Data are represented as mean ± SEM. Significance was determined by Student's *t* test analysis, *P < 0.05. Source data are provided as a Source Data file

**Adipose-specific *HuR* deletion exacerbates insulin resistance**. Obesity is closely related to glucose intolerance and insulin resistance. Compared with NCD-fed CTR mice, HFD-fed CTR mice showed impaired glucose tolerance and insulin resistance (Supplementary Fig. 3c, d), accompanied with a decreased adipose HuR expression. Then, we assessed the effect of adipose-specific *HuR* deletion on glucose homeostasis and insulin sensitivity. HFD-fed HuR^AKO mice exhibited glucose intolerance and insulin resistance as compared with control mice (Fig. 6a, b). To understand the mechanism by which *HuR* ablation damaged insulin tolerance, we further examined the phosphorylation of insulin-stimulated Akt Ser473 in epiWAT, liver and skeletal muscle of mice. Insulin-induced Akt phosphorylation was attenuated in adipose tissue of HuR^AKO mice, indicating that *HuR* ablation in adipocytes could exacerbate HFD-induced insulin

resistance in adipose tissue. However, given a significant reduction of basal Akt signaling and an increased fold under insulin-induced situation in liver and muscle tissues, adipose-specific *HuR* depletion led to their increased sensitivity to insulin under HFD (Fig. 6c, d). Consistently, HFD-fed HuR^AKO mice showed an increased fasting insulin level and decreased serum adiponectin level as compared with controls (Fig. 6e, f). Thus, adipose-specific *HuR* deletion exacerbates HFD-induced insulin resistance.

**HFD-fed HuR^AKO mice show increased hepatic steatosis**. Given that hepatic steatosis is closely associated with obesity and insulin resistance, we assessed the impact of adipose *HuR* deletion on hepatic lipid deposition. HFD-fed CTR mice showed more lipid

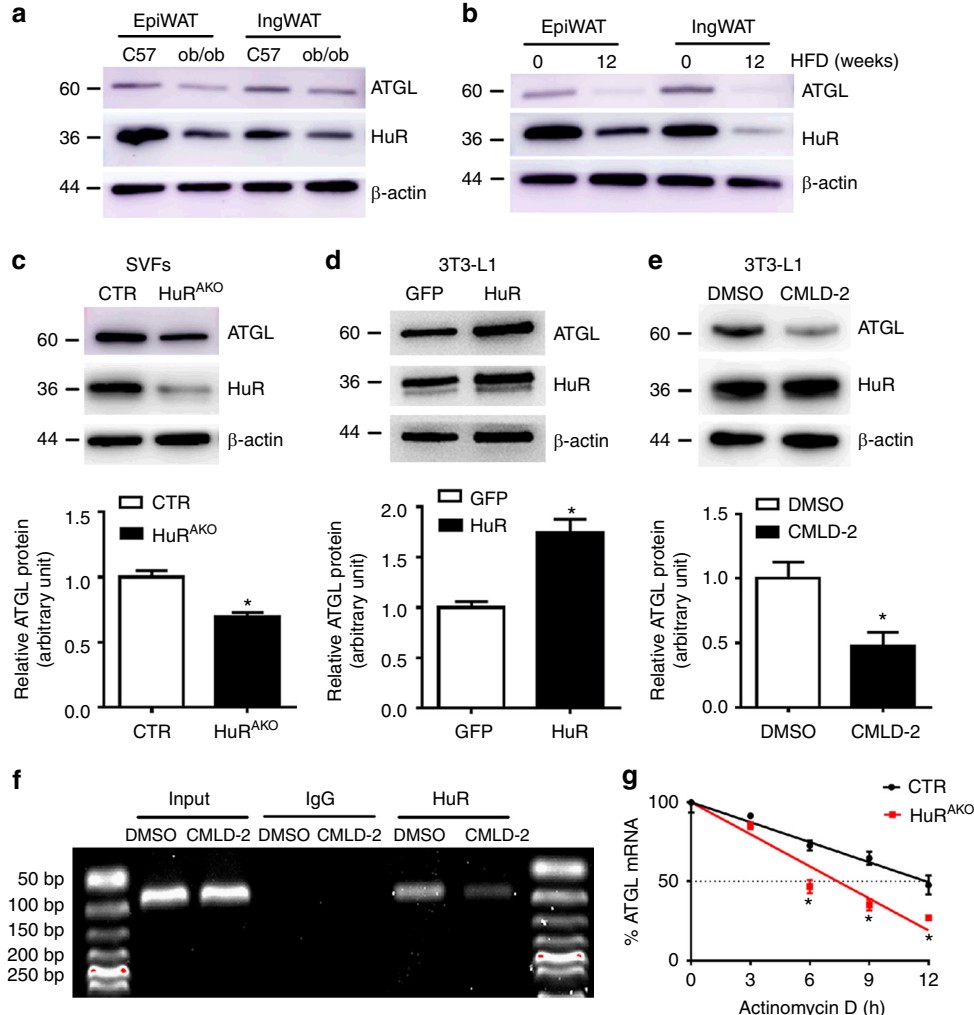

**Fig. 5** HuR regulates *ATGL* mRNA stability. **a** Western blot analysis of HuR and ATGL expression in adipose tissue (epiWAT and ingWAT) of 20-week-old C57BL/6J and ob/ob mice and quantification ($n = 3$), *comparison of ob/ob vs. C57. **b** Eight-week-old male C57BL/6J mice were fed with an HFD for an additional 12 weeks; western blot analysis of HuR and ATGL in adipose tissue and quantification ($n = 3$), *comparison of HFD (12w) vs. HFD (0w). **c** SVFs were isolated from control and HuR$^{AKO}$ mice and differentiated to adipocytes. Western blot analysis of ATGL and quantification ($n = 3$), *comparison of HuR$^{AKO}$ vs. control. **d** Differentiated 3T3-L1 adipocytes were infected with adenovirus expressing GFP or HuR for 48 h. Western blot analysis of ATGL protein level and quantification ($n = 3$), *comparison of HuR vs. GFP. **e** 3T3-L1 adipocytes were stimulated with CMLD-2 (30 µM) for 24 h. Western blot analysis of ATGL protein level and quantification ($n = 3$), *comparison of CMLD-2 vs. DMSO. **f** Differentiated 3T3-L1 adipocytes were stimulated with CMLD-2 (30 µM) or DMSO for 24 h. RNA immunoprecipitation with anti-HuR antibody or control IgG. **g** SVFs isolated from control and HuR$^{AKO}$ mice were differentiated to adipocytes and stimulated with 10 µg mL$^{-1}$ actinomycin D. *ATGL* mRNA level was determined by qPCR ($n = 4$), *comparison of HuR$^{AKO}$ vs. control. Data are represented as mean ± SEM. Significance was determined by Student's *t* test analysis, *$P < 0.05$. Source data are provided as a Source Data file

accumulation in the liver than NCD-fed CTR mice and less lipid accumulation than HuR$^{AKO}$ mice under an HFD diet (Fig. 7a). Furthermore, liver triglycerides and cholesterol levels were significantly higher in HuR$^{AKO}$ mice relative to controls (Fig. 7b). The mRNA levels of sterol regulatory element-binding protein-1c (*Srebp-1c*), fatty acid transport protein 1 (*Fatp1*), *Fatp2*, *Fatp3*, *Fatp4* and fatty acid synthesis genes, such as fatty acid synthase (*FAS*) and *ACC* were significantly increased in livers of HuR$^{AKO}$ mice, but *ATGL* expression were changed only slightly (Fig. 7c). Altogether, these data suggest that loss of adipocyte HuR exacerbated HFD-induced liver steatosis in part due to increased fatty acid uptake and lipogenesis.

Obesity is often accompanied by macrophage infiltration, which facilitates chronic inflammation and insulin resistance. Immunofluorescent staining of HuR and macrophage marker F4/80 suggested that macrophages are the primary source of HuR in adipose tissue from HuR$^{AKO}$ mice and revealed increased crown-like structures in epididymal adipose tissue of HFD-fed HuR$^{AKO}$ mice (Fig. 7d). In addition, mRNA levels of *F4/80* and inflammatory genes such as *CD68*, monocyte chemoattractant protein 1 (*MCP-1*), tumor necrosis factor-α (*TNF-α*) and interferon-γ (*IFN-γ*) were upregulated in HuR$^{AKO}$ adipose tissue (Fig. 7e). Thus, adipose-specific *HuR* deficiency may cause increased inflammation in adipose tissue.

**Reduced *HuR* and *ATGL* expression in obese patients**. Finally, we detected whether the expression of HuR and ATGL was changed in adipose tissues of obese patients. The protein level of HuR was significantly suppressed in subcutaneous adipose tissue of obese patients, with a parallel decrease in ATGL expression (Fig. 7f), consistent with observations in HuR$^{AKO}$ mice. More

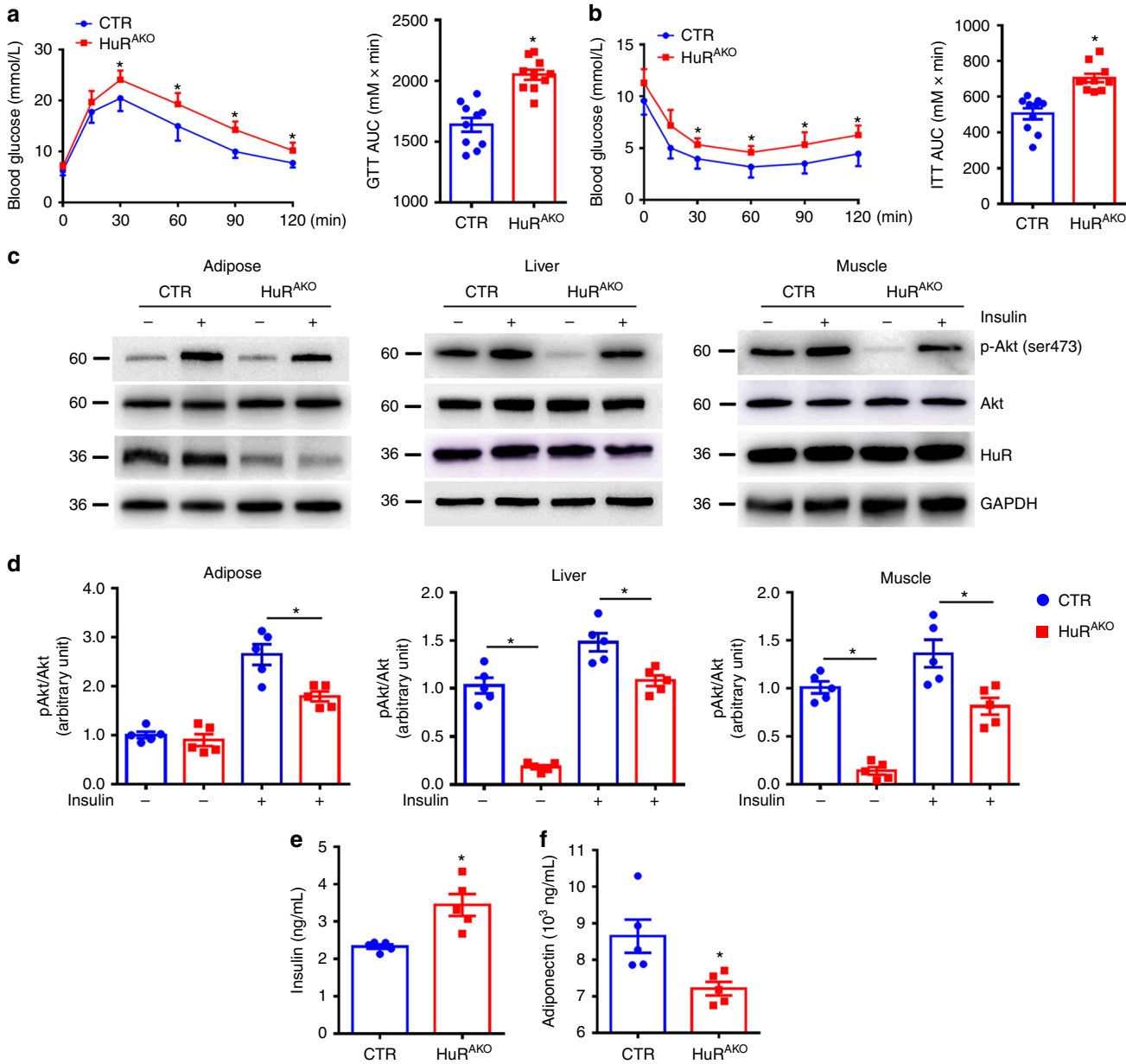

**Fig. 6** Adipose-specific *HuR* deletion exacerbates insulin resistance. **a**, **b** Glucose tolerance test (**a**) and insulin tolerance test (**b**) in control and HuR[AKO] mice fed an HFD ($n = 10$), *comparison of HuR[AKO] vs. control. Right panel, area under curve. **c**, **d** Western blot analysis of Akt phosphorylation and HuR in adipose tissue, liver and muscle of control and HuR[AKO] mice with HFD. Protein levels were normalized to GAPDH level ($n = 5$), *comparison of HuR[AKO] vs. control. **e**, **f** Serum insulin and adiponectin levels in overnight-fasted control and HuR[AKO] mice fed an HFD ($n = 5$), *comparison of HuR[AKO] vs. control. Data are represented as mean ± SEM. Significance was determined by Student's *t* test analysis, *$P < 0.05$. Source data are provided as a Source Data file

importantly, we discovered the tendency quantitatively with a regression line (Fig. 7g), which indicated that the normalized protein expression of HuR was reversely correlated with body mass index (BMI). Together, these results demonstrate that HuR might have a regulatory role in ATGL expression and obesity in humans.

## Discussion

In this study, we explored the effect of adipose-specific *HuR* deletion on obesity and related metabolic disturbances in mice. HuR[AKO] mice were more susceptible to the development of HFD-induced obesity via adipocyte hypertrophy, which might be attributed to impaired lipolysis. In addition, HFD-fed HuR[AKO] mice showed insulin resistance and exacerbated hepatic steatosis.

Mechanistically, we demonstrated that the regulation of lipolysis by HuR was mediated by ATGL. HuR could bind to the 3′UTR of *ATGL* mRNA and increased its stability (Fig. 7h). Taken together, our findings establish HuR as a regulator of ATGL-mediated lipolysis.

HuR, as a member of RNA-binding proteins, has been reported to regulate the expression of many molecules by a post-transcriptional mechanism, which indicates its multifunctionality. In our recent study, HuR could regulate the stability of *GTP cyclohydrolase 1* mRNA in endothelial cells[31]. HuR is involved in many biological processes, including carcinogenesis, inflammation, cell survival and apoptosis[24,26,27,32–34]. However, the in vivo functions of HuR have not been clearly elucidated due to its critical role in embryonic development and the embryonic lethal phenotype of global *HuR*-knockout mice[28]. Recent studies

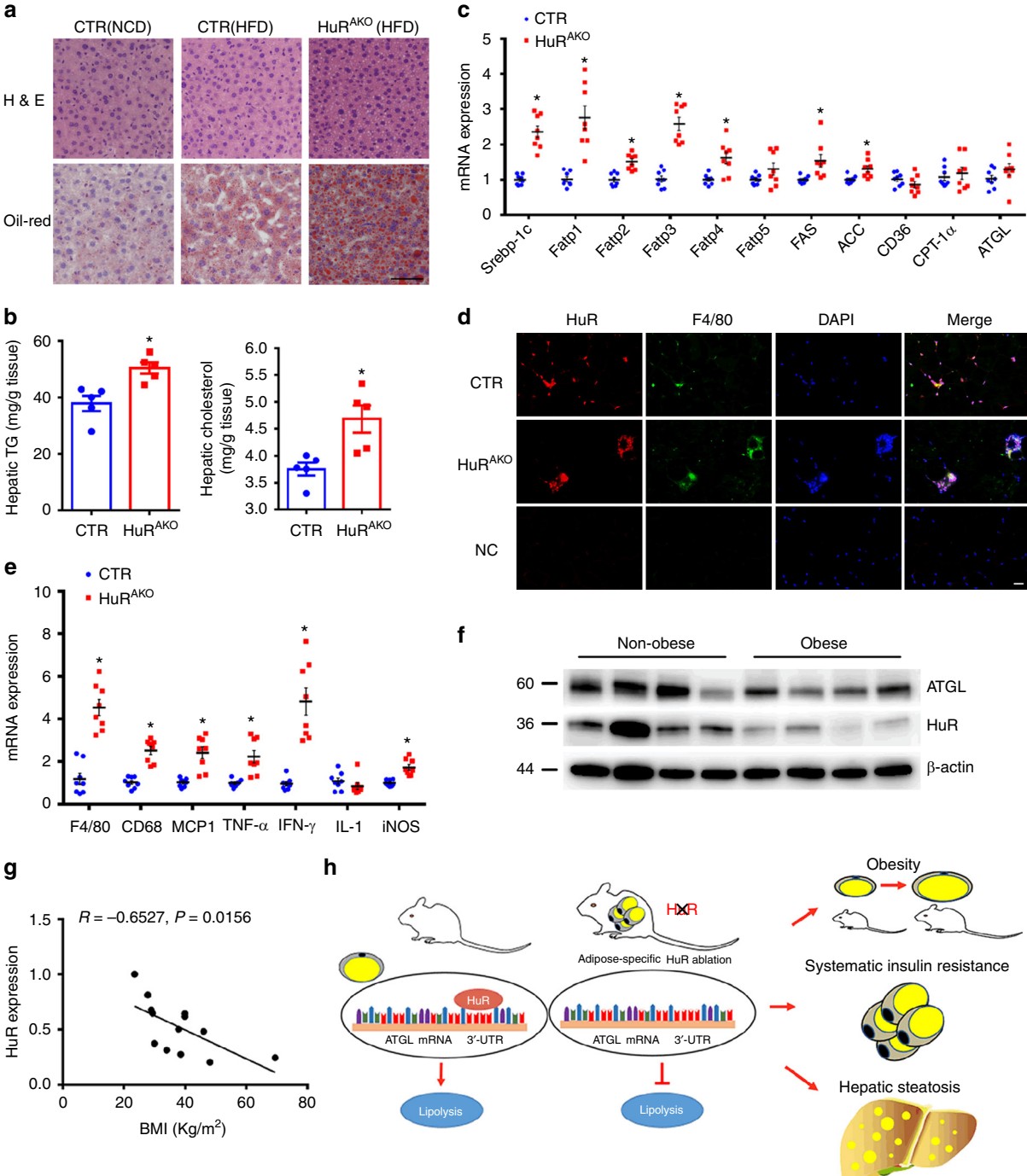

**Fig. 7** HFD-fed HuR^AKO mice show increased hepatosis. **a** Representative images of H&E- and Oil-red O-stained sections of liver tissues from NCD-induced control, HFD-induced control and HuR^AKO mice. Scale bar: 50 μm. **b** Levels of hepatic cholesterol and triglycerides in livers of control and HuR^AKO mice fed an HFD (*n* = 5), *comparison of HuR^AKO vs. control. **c** qPCR analysis of mRNA levels of lipogenesis-related genes in liver from control and HuR^AKO mice fed an HFD (*n* = 8), *comparison of HuR^AKO vs. control. **d** Immunofluorescence staining of HuR and F4/80 in adipose tissue. DAPI: nucleus. Scale bar: 20 μm. **e** qPCR analysis of mRNA levels of inflammation-related genes in livers from control and HuR^AKO mice fed an HFD (*n* = 8), *comparison of HuR^AKO vs. control. **f** Western blot analysis of ATGL and HuR protein levels in adipose tissues from normal and obese patients (*n* = 4 for normal or obese individuals). **g** Correlation of normalized HuR protein expression and BMI in adipose tissues of obese patients (total 13 samples). **h** Schematic diagram about the mechanism of HuR in adipose tissue and obesity. Data are represented as mean ± SEM. Significance was determined by Student's *t* test analysis and Pearson's correlations, *P < 0.05. Source data are provided as a Source Data file

reported that intestinal-specific *HuR* deletion reduced the tumor burden in models of intestinal tumorigenesis and inflammatory colon carcinogenesis[29]. B lineage-specific deletion of *HuR* impaired the survival of B cells in bone marrow and humoral immunity[30]. Here we generated adipose-specific *HuR*-knockout

mice and found that HuR could protect against HFD-induced obesity and insulin resistance, which amplifies our understanding of the functions of *HuR*. Moreover, HuR function is regulated via changes in its subcellular localization, affinity for RNA, abundance, and cleavage, which in turn influence its ability to affect

the fate of target mRNAs[35–38]. In this study, the mRNA levels of *HuR* in epiWAT, subWAT and BAT were decreased from HFD-fed mice (Supplementary Fig. 1b), which suggested that HFD-induced repression of HuR could be attributed to its mRNA reduction. High-fat diet may either interfere with the transcription of *HuR* or, alternatively, increase the decay of *HuR* mRNA. This important issue needs to be addressed by future studies.

ATGL is the key lipase that hydrolyzes triglycerides into diglycerides, which are further catalyzed by hormone-sensitive lipase (HSL) into monoglycerides and glycerols[39]. Given its critical role in lipolysis and metabolic homeostasis, the regulation of ATGL activity and expression need exploration. ATGL activity can be modified by its interaction with other proteins such as G0/G1 switch gene 2, which inhibits ATGL action by physical interaction of its N-terminal with the N-terminal patatin domain of ATGL[40]. The activity of ATGL lipase is also regulated by different phosphorylation sites[15,41,42]. In addition, several factors have been identified to regulate *ATGL* expression[15–19]. PPARγ can directly regulate *ATGL* expression in adipocytes in vitro and in vivo. Treatment with rosiglitazone, a PPARγ agonist, and thiazolidinedione, an antidiabetic agent, increases *ATGL* mRNA and protein expression in WAT and BAT of mice under the background of HFD or leptin deficiency[18]. mTORC1 suppressed lipolysis and promoted fat accumulation in mammalian cells primarily by inhibiting transcription of *ATGL*, whereas rapamycin, an mTORC1 inhibitor, reversed the loss of ATGL[17]. In this study, we have demonstrated that HuR, as an mRNA-binding protein, can target *ATGL* to modulate its mRNA stability and lipase expression positively, which reveals a potential mechanism in regulating lipolysis.

Since the triglycerides within adipocytes constitute >90% of adipocyte volume and it is the major volume component of the adipose mass[43], suppressed lipolysis could cause retained lipids in adipocytes and thereby expand adipose tissue. So far, there is no consensus of opinion about the relationship between ATGL and insulin sensitivity. Recent studies reported that adipose overexpression of ATGL attenuated diet-induced obesity and improved insulin sensitivity[44]. Adipose-specific knockout of sirtuin 6 (Sirt6) in mice exacerbated insulin resistance and inflammation in HFD-induced obesity, in which ATGL expression and lipolysis were inhibited[39], which was consistent with our study. Thus, suppressed ATGL levels due to *HuR* knockout may contribute to reduced lipolysis and the phenotype with obesity in the HuR^AKO mice. Emerging evidence suggests that impaired adipocyte lipolysis causes the proinflammatory immune cell infiltration of metabolic tissues in obesity, which contributes to the development and exacerbation of insulin resistance[39,45]. In our study, *HuR* deletion deteriorated insulin resistance and inflammation infiltration in adipose tissue. Also, obese HuR^AKO mice showed severe hepatic steatosis, which was marked by excess triglycerides accumulation and increased expression of genes involved in fatty acid uptake and lipogenesis. Thus, the HuR-ATGL-lipolysis axis may be required for maintaining normal adipocyte size and adiposity. Inhibited lipolysis in HuR^AKO mice caused marked adipocyte hypertrophy, which responded poorly to insulin treatment. The elevated inflammation levels also contributed to insulin resistance in HuR^AKO mice.

In summary, we demonstrate that HuR functions as a positive regulator of lipolysis by targeting *ATGL* to maintain its mRNA stability. Adipose-specific deletion of *HuR* promoted HFD-induced obesity, impaired adipose function and deteriorated glucose intolerance and insulin resistance. HuR may be a potential therapeutic target for ameliorating obesity and obesity-related metabolic disorders.

## Methods

**Animals**. *HuR^flox/flox* (#021431)[46] and *AdipoQ-Cre* mice (#024671)[47] were purchased from Jackson laboratories (Bar Harbor, ME). Ob/ob and C57BL/6J mice were purchased from Beijing Vital River Laboratory Animal Technology Co. HuR^AKO mice were generated by crossbreeding *HuR^flox/flox* mice with *AdipoQ-Cre* mice. *HuR^flox/flox:AdipoQ-Cre−* mice were used as controls. Starting at 8 weeks of age, mice were fed an HFD (D12492; Research Diets) consisting of 60% fat or normal chow diet (D12450B; Research Diets) and body weight was monitored weekly. Mice were housed on a 12-h light/dark cycle and given ad libitum access to food and water. All animal protocols were approved by the Institutional Animal Care and Use Committee of Cheeloo College of Medicine, Shandong University. All relevant ethical regulations were adhered to.

**Stromal vascular fraction isolation**. Mice at age 6−8 weeks were sacrificed and epididymal adipose tissues were removed, minced in phosphate-buffered saline (PBS) containing 10 mM CaCl₂, and digested at 37 °C for 60 min in DMEM containing 10 mM CaCl₂, 1.5 units ml⁻¹ collagenase D (Sigma, 11088866001), and 2.4 units ml⁻¹ dispase II (Sigma, D4693). Tissue suspensions were filtered through a 100-μm filter and centrifuged at 600 × g for 5 min. Primary adipocytes (top floating fractions) and SVFs (pellets) were collected. SVFs were resuspended and filtered through a 40-μm filter, and then grown in DMEM containing 25 mM glucose and 10% fetal bovine serum (FBS).

**Preadipocytes differentiation**. Mouse 3T3-L1 cells were purchased from ATCC and grown in DMEM supplemented with 10% FBS. After 3T3-L1 cells or SVFs reached confluence (day 0), differentiation of the cells was induced in DMEM containing 10% FBS, methylisobutylxanthine (520 μmol L⁻¹), dexamethasone (1 μmol L⁻¹), and insulin (167 nmol L⁻¹) for 48 h. From day 3, the culture medium was replaced on alternate days with DMEM containing 10% FBS and 167 nmol L⁻¹ insulin. At day 8 of differentiation, cells were ready for analysis.

**White adipose tissue and liver histology analysis**. Adipose tissue and livers were isolated and fixed in 4% formaldehyde and maintained at 4 °C until use. The fixed tissues were dehydrated and processed for paraffin embedding, and 5-μm sections were stained with hematoxylin and eosin (H&E). Adipocyte size was determined by using ImageJ (US National Institutes of Health), measuring a minimum of 300 cells per group. Immunofluorscence and immunohistochemical stainings were performed with HuR antibody (Millipore, #07-468, 1:200) and F4/80 antibody (abcam, ab16911, 1:50). Livers were isolated and frozen in OCT embedding medium; 10-μm sections were used to stain with Oil-red O.

**Metabolic cage**. Food intake, O₂ consumption, RER, heat production and physical activity were determined by using a TSE PhenoMaster/LabMaster system (Germany). Experiments involved control and HuR^AKO mice at 14–16 weeks old. Mice were individually monitored for 48 h and data were collected in a time interval of 30 min after 1-day adaption.

**Lipolysis assay**. WAT was minced into small pieces (1–2 cm²) in cold PBS. Fat explants (5–6 pieces) were incubated at 37 °C for 2 h in lipolysis buffer and then stimulated with isoproterenol (ISO, 1 μM). Differentiated 3T3-L1 adipocytes was transfected with adenovirus expressing HuR or GFP (Shanghai Genechem Co., Ltd.) and stimulated with ISO (10 μM). SVF-derived adipocytes were stimulated with ISO (10 μM). HFD-fed mice were injected with CL316243 (Sigma, C5976) (1 mg kg⁻¹ body weight) and angular vein blood was collected after 0 and 15 min. FFA (abcam, ab65341) and glycerol (explant for zenbio, LIP-6-NC; cell culture for abcam, ab185433) were measured and normalized to total adipocyte protein levels as the index of lipolysis.

**Lipid profile assays**. Mice were fasted overnight and angular vein blood was collected. Serum samples were stored at −80 °C until use. Serum levels of insulin and adiponectin were measured by using ELISA kits (Insulin for Millipore EZRMI-13K; adiponectin for R& D MRP300). Concentrations of TG, cholesterol, HDL, LDL were measured by assay kits. Liver samples were homogenized in 100% ethanol and centrifuged. The upper aqueous fractions were used to measure TG and cholesterol (Jiancheng Bioengineering Institute, Nanjing, China).

**Metabolic studies**. For glucose tolerance test, mice were fasted overnight and then received an intraperitoneal (i.p.) injection of glucose (0.75 g kg⁻¹ body weight). For insulin tolerance test, mice were fasted for 4 h before receiving an i.p. injection of insulin (1.5 U kg⁻¹ body weight). Blood glucose concentrations were measured at 0, 15, 30, 60, 90 and 120 min after glucose or insulin injection. Acute insulin challenge experiments were performed on anesthetized mice fasted for 4 h. At 20 min after an i.p. injection with insulin (2 U kg⁻¹ body weight), the remaining liver, muscle, and fat were snap-frozen for subsequent protein extraction.

**RNA-immunoprecipitation (IP) assay**. The Magna RIP kit was used for RNA IP assay. The differentiated SVFs were treated with CMLD-2 (30 μM, Millipore,

538339) or DMSO for 24 h. Briefly, whole-cell lysates were incubated at 4 °C overnight with magnetic protein A/G beads pretreated with 5 µg rabbit IgG or HuR antibody (Millipore, #07-468). Beads were washed and incubated with proteinase K buffer (30 min at 55 °C), then RNA was isolated from immunoprecipitates, and cDNA was synthesized. The primers for PCR are shown in Supplementary Table 1.

**Quantitative PCR (qPCR).** Total RNA was extracted from WAT or liver by using TRIzol reagent (Invitrogen, Carlsbad, CA). Total RNA was extracted from differentiated SVFs by using the RNeasy Lipid Tissue Mini kit (Qiagen, Valencia, CA, USA). An amount of 1 µg RNA was reverse-transcribed into cDNA by using the PrimeScript RT Reagent Kit (Takara Biomedical Technology). PCR amplification involved using the SYBR PCR mix (Takara Biomedical Technology Co.). The primers for quantitative PCR are listed in Supplementary Table 1.

**Western blot analysis.** Briefly, 20 µg total lysates from tissues or cells were run on a 10% SDS-PAGE gel and immunoblotted with the primary antibodies (1:1000) to HuR (Millipore, #07-468), ATGL (abcam, ab109251), Perilipin-1 (abcam, ab172907), HSL(cst, 4107), Akt (cst, 4691), p-Akt (ser473, cst, 4060), ADRB1 (abcam, ab3442), ADRB2 (abcam, ab182136), ADRB3 (abclonal, A8607), β-actin (Proteintech, 66009-1), GAPDH (Proteintech, 60004-1). The intensity of bands was measured by using ImageJ. All experiments were repeated at least three times and mean values were derived. All the uncropped blots are included in the Source Data file.

**RNA-seq analysis.** Total RNA was extracted from control and HuR-deficient adipose tissues. The transcriptome sequencing experiments were performed by KangCheng Bio-tech Company (Shanghai, China). The transcriptome library for sequencing was generated using KAPA-Stranded RNA-Seq Library Prep Kit (Illumina) following the manufacturer's recommendations. The clustering of the index-coded samples was used KAPA RNA Adapters set1/set2 for Illumina. After clustering, the libraries were sequenced on Illumina Hiseq X Ten platform using (2 × 150 bp) paired-end module. The differentially expressed genes were identified with $P$ value < 0.05 and a fold-change of >1.5 between two groups.

**Human subcutaneous adipose tissue.** Biopsies of subcutaneous adipose tissue were obtained from 25 Chinese people who were undergoing elective surgery in Qilu Hospital of Shandong University. All subjects provided their written informed consent. All procedures that involved human samples were approved by the Ethics Committee of Qilu Hospital of Shandong University. All relevant ethical regulations were followed. Obese individuals were defined as those with a BMI ≥ 30 kg m$^{-2}$.

**Statistical analysis.** All the replicate experiments (including cell and mouse-based experiments) are biological replicates, which were repeated at least three times. All analyses involved use of SPSS v23 (SPSS Inc., Chicago, IL). All data are represented as mean ± SEM. Comparison of two groups was determined by Student's $t$ test and multiple groups by one-way ANOVA with Tukey's post-hoc test. The assumption of normality was tested by the Shapiro−Wilks test. All statistical tests were two-tailed, and $P < 0.05$ was considered statistically significant.

**Reporting summary.** Further information on research design is available in the Nature Research Reporting Summary linked to this article.

## Data availability

Gene expression RNA-seq data have been deposited at the NCBI Gene Expression Omnibus (GEO). The accession number is GSE130147. Data supporting the findings of this study are available within the article and its Supplementary Information files. All relevant data are available from the authors on reasonable request.

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

## Acknowledgements

This study was supported by grants from the National Natural Science Foundation of China (No. 81770473, 81570393, 81570324, 81425004, 81770442), the Taishan Scholar Project of Shandong Province of China (No. tsqn20161066), the National Key Project of Chronic Non-Communicable Disease of China (No. 2016YFC1300403) and the China Postdoctoral Science Foundation (No.2019M652410).

## Author contributions

J.L., L.G., S.L., Y.Z., C.Z. and M.T. designed and performed the research, H.L., P.B., J.Y., C.O., X.J. and J.W. analyzed data, Y.Z., Q.M., C.Z. and W.Z. conceived the project, reviewed the data and wrote the manuscript.
