## [Peer Review File · Nature Communications]

Reviewers' comments:

Reviewer #1 (Remarks to the Author):

The manuscript entitled "Adipose HuR protects against diet-induced obesity and insulin resistance" describes the functional role of HuR in obesity and insulin resistance by use of an adipose-specific HuR knockout approach. The authors convincingly demonstrate that adipose-specific depletion of HuR leads to adipocyte hypertrophy without affecting adipocyte differentiation. Furthermore, they show that adipose-specific HuR ablation has a significant effect on diverse key metabolic functions e.g. an increase in lipolysis, glucose intolerance, hypercholesterolemia and adipose tissue inflammation along with a decreased insulin resistance. By using a gene array the authors have identified a large target number of genes under the potential control of HuR (103 negatively regulated and 113 positively regulated by HuR). Importantly, some lipolysis-associated genes including ATGL and PLIN were significantly reduced in HuR AKO mice. By employing actinomycin-D and RNA-pulldown assays, the authors furthermore show that ATGL, the rate limiting enzyme of adipocyte lipolysis, is a direct target of HuR dependent mRNA-stabilization and which is primarily responsible for the observed obese phenotype. Finally, they demonstrate reduced HuR and ATGL levels in adipose tissue from obese patients when compared to tissue from healthy volunteers. From these data they suggest that HuR is a critical positive regulator of ATGL expression and lipolysis which can protect against obesity and metabolic syndromes.

To my opinion, this is a comprehensive and interesting study supporting novel information about the physiological role of HuR in obesity and metabolic syndromes. For this reason, I guess that the study has a strong impact for a large readership of the scientific community. Although the overall quality and set-up of most experiments is high and well-conceived, there are some critical aspects which need to be addressed in a revised manuscript. My recommendation: "Accept after major revisions"

More specifically:

- # 1. Figure 1A/B. Please, show mRNA levels of ATGL and HuR to see whether the inhibited expression is also found on the mRNA level.
- # 2. Fig. 1D/E: The adipose-specific knockout of HuR is not complete neither at the mRNA (panel D) nor on the protein level (panel E). I guess that this is due to a contamination with adjacent tissue or infiltrating immune cells. Therefore, the authors should prove the knockout efficacy in adipose tissue additionally by IHC (supplementary to data which are presented in Fig. 3A).
- # 3. Fig. 4A. Since β -adrenergic receptors can be target of 3' UTR-driven mRNA-stability, the reduction in ISO-stimulated glycerol and FFA release could, at least partially, be due to HuR depletion-dependent reduction in β -receptor expression. The levels of β receptor expression in EpiWAT explants from wild-type and HuR AKO mice should be shown to test the possible impact of reduced β -receptor expression.
- # 4. Fig. 5F: The authors should prove inhibitory effects of HuR binding to ATGL mRNA by CMLD-2 by RNA pull-down assay. Please give information in the figure legend which cells were used for RIP assays.
- # 5. Fig. 5G: it would be interesting to know whether HFD would exert a similar negative effect on ATGL mRNA-stability as genetic HuR depletion.
- # 6. Fig. 6C: please include Westernblots showing HuR abundance. The reduction in p-Akt levels in liver in HuR AKO mice is not convincing since the levels of total Akt are also reduced by insulin. Please show another blot which represents the graphic data in panel D.
- # 7. Fig. 7A. Please show also IHC data from CTR animals under a normal diet.
- # 8. Fig. 7D. Please show also IHC data after staining of HuR to test whether macrophages are the primary source of HuR in adipose tissue from HuR knock outs (see also comment # 2).

8. Legend of Figs 7F,G. Does the n number (6) correspond to the total number of patients ?. In the latter case, a statistic analysis to my opinion is not valid since the number of non-obese controls was only 2 (according to what is shown in the W.blot in panel F).

Conceptual comments:

1. According to results from HuR ko. mice, the authors conclude that HuR has a protective role against high-fat diet-induced obesity and this statement is also declared in the title of the ms. Besides of the genetic model of obesity (ob/ob mice), most of experiments in which the functional impact of HuR was assessed were performed under conditions of a high fat diet (HFD). Thereby, according to data presented in Fig. 1B and Fig. 5B, HFD has a strong suppressive effect on HuR and ATGL abundance similar to genetic depletion of HuR which seems physiologically relevant to avoid from generation of increased free fatty acids as energy substrate (shown in Fig. 1E). Nevertheless, if HuR exerts an overall protective role in diet-induced obesity, I am wondering why it is downregulated instead of being upregulated? Since most of their conclusions rely on data obtained with HuR knock-out mice under a high fat diet, they should additionally show the effects of HFD in wild-type animals. Or in other words: They show pathophysiological effects of a gene knockout under experimental conditions (HFD) which per se induce a strong ablation of the gene. The authors should comment on this phenomenon in the discussion.

2. Since the mRNA and protein of HuR are quite stable (due to HuR-triggered autoregulation), an important issue which is not addressed in this ms. is the mechanism which is underlying HFD-induced HuR repression. From the literature, HuR is known to be mainly regulated by means of intracellular localization and modulation of RNA-binding with both of these modes are controlled by diverse posttranslational modifications. The authors should at least discuss possible mechanisms which could explain the HFD-induced repression of HuR e.g. transcription, translation, reduction in mRNA or protein half-life, ubiquitin-dependent proteasomal or caspase-triggered degradation (see also comment # 1).

Minor:

1. P. 8 (line 162). Please replace "modification" by "modulation". The term modification is generally used for post-translational events.
2. Material and Methods: The authors should mention the source of CMLD-2. To the best of my knowledge this inhibitor is no longer commercially available.
3. P12, line 245: "hormone-sensitive" instead of "sensitive"

Reviewer #2 (Remarks to the Author):

In this paper, authors demonstrate that the adipose-specific deletion of HuR is associated with obesity and dysregulated fatty acid metabolism in adipose tissue. HuR is also downregulated in commonly used mouse models of obesity (ob/ob mice and HFD-fed C57Bl/6) and human obesity. HuR-deficiency downregulated the expression of key proteins of lipolysis including ATGL and PLIN. Authors convincingly demonstrate that ATGL mRNA stability is regulated by HuR. Furthermore, HuR-deficiency was associated with reduced serum fatty acid and glycerol levels, and exacerbated HFD-induced insulin resistance and liver steatosis.

In principal, the paper is interesting showing that HuR-affects adipose tissue function and metabolic disease. However, it is important to note that reduced ATGL expression in adipose tissue rather counteracts insulin resistance and liver steatosis. Thus, the phenotype of adipose-specific HuR-deficiency cannot be explained by reduced lipolysis which should be discussed in the manuscript. Accordingly, authors show that many genes are up- or downregulated in adipose tissue. Which of these genes are predicted targets of HuR and which of them affect adipose tissue function?

PLIN plays a key role in the regulation of lipolysis and triglyceride storage, and is strongly downregulated. Is it a HuR-target?

Minor comments:

11: respiration exchange rate/ respiratory exchange rate

Fig.3: Quantify changes in oxygen consumption, respiratory exchange ratio and heat production.

124: lipoapoprotein lipase/lipoprotein lipase

124: "β-oxidative genes such as acetyl-CoA carboxylase (ACC)". ACC is involved in fatty acid synthesis.

Fig.4 Is differentiation unchanged in SVF lacking HuR and 3T3L1 overexpressing HuR?

Fig. 7 is there a correlation between BMI with HuR expression?

279: Which experiments support the hypothesis that "HuR may be a potential therapeutic target for ameliorating obesity and obesity-related metabolic disorders".

Dear Editors and Reviewers:

Thank you for your letter and for the reviewers' comments concerning our manuscript entitled "**Adipose HuR protects against diet-induced obesity and insulin resistance**" (ID: NCOMMS-18-30365). We thank the editors and the reviewers for their careful readings. We are delighted to know that "***this is a comprehensive and interesting study supporting novel information about the physiological role of HuR in obesity and metabolic syndromes. For this reason, I guess that the study has a strong impact for a large readership of the scientific community***" "***the overall quality and set-up of most experiments is high and well-conceived***" (reviewer #1), and "***the paper is interesting showing that HuR-affects adipose tissue function and metabolic disease***" (reviewer #2). Thanks to these positive comments, we are extremely encouraged to revise the manuscript by performing all additional experiments. We found the reviewers' comments very helpful, which guided our revisions resulting in this much improved manuscript. Because there are limited changes to this paper based on the review, changes in text are noted by highlights.

Responds to the reviewer's comments:

Reviewer #1:

1. "*Figure 1A/B. Please, show mRNA levels of ATGL and HuR to see whether the inhibited expression is also found on the mRNA level.*"

Response: Thanks for this suggestion. We measured the mRNA levels of ATGL and HuR of adipose tissues by RT-PCR, which showed that the mRNA levels of HuR and ATGL were decreased in those two obesity models (**Supplementary Figure 1A-1D**).

We have corrected the following sentence in Page 4 and 8 in the revised text.

“The protein and mRNA levels of HuR were significantly decreased in WAT and BAT from the leptin mutant (ob/ob) and HFD-fed mice, the models of obesity and type 2 diabetes, as compared with their controls (**Figure 1A-1B and Supplementary Figure 1A-1B**).”

“The protein and mRNA levels of ATGL were decreased in these two obesity models, which were consistent with those of HuR (**Figure 5A-5B and Figure 1C-1D**).”

2. *“Fig. 1D/E: The adipose-specific knockout of HuR is not complete neither at the mRNA (panel D) nor on the protein level (panel E). I guess that this is due to a contamination with adjacent tissue or infiltrating immune cells. Therefore, the authors should prove the knockout efficacy in adipose tissue additionally by IHC (supplementary to data which are presented in Fig. 3A).”*

Response: Many thanks for this important suggestion. We examined the expression of HuR in epiWAT, subWAT and BAT by immunohistochemistry assay which showed that HuR was expressed in macrophage and other non-adipocytes, but not expressed in adipocytes from HuR^{AKO} mice (**Supplementary Figure 1E and Figure 7D**).

We have added the following sentence in Page 4.

“HuR was expressed in macrophage and other non-adipocytes of adipose tissues from HuR^{AKO} mice (**Supplementary Figure 1E**).”

3. *“Fig. 4A. Since β -adrenergic receptors can be target of 3'UTR-driven mRNA-stability, the reduction in ISO-stimulated glycerol and FFA release could, at least partially, be due to HuR depletion-dependent reduction in β -receptor expression. The levels of β receptor expression in EpiWAT explants from wild-type and HuR AKO mice should be shown to test the possible impact of reduced β -receptor expression.”*

Response: Thanks for this question. We detected the β receptors expression

in epiWAT explants. And we found that there were no difference in levels of β adrenergic receptors, including β 1 adrenergic receptor (ADRB1), β 2 adrenergic receptor (ADRB2) and β 3 adrenergic receptor (ADRB3), between CTR and HuR^{AKO} mice (**Supplementary Figure 1F**). The β adrenergic receptor may not be the target of HuR.

4. *“Fig. 5F: The authors should prove inhibitory effects of HuR binding to ATGL mRNA by CMLD-2 by RNA pull-down assay. Please give information in the figure legend which cells were used for RIP assays.”*

Response: Thanks for this suggestion. We performed the RNA-IP assay to examine the inhibitory effect of HuR binding to ATGL mRNA by CMLD-2 and replaced with new **Figure 5F**. For RIP assay, differentiated SVF cells were used and we have added this information in the figure legend.

We have added the following sentence in Page 9.

“which was inhibited by CMLD-2 (**Figure 5F**).”

5. *“Fig. 5G: it would be interesting to know whether HFD would exert a similar negative effect on ATGL mRNA-stability as genetic HuR depletion.”*

Response: Thank you for the interesting question. We cultured the epididymal adipose tissues from the HFD or NCD-fed mice and treated with actinomycin D followed by ATGL mRNA measurement. The half-life of ATGL mRNA in HFD mice (about 8h) was much shorter than that in NCD mice (about 10h) (**Supplementary Figure 2C**), which was consistent with the down-regulation of HuR under high fat diet.

We have added the following sentence in Page 9.

“Moreover, in consistent with the down-regulation of HuR under high fat diet, the ATGL mRNA stability in the epididymal adipose tissue from HFD-fed mice was decreased compared with that from NCD-fed mice (**Supplementary Figure 2C**)”

6. *“Fig. 6C: please include Western blots showing HuR abundance. The reduction in p-Akt levels in liver in HuR AKO mice is not convincing since the levels of total Akt are also reduced by insulin. Please show another blot which represents the graphic data in panel D.”*

Response: Thanks for your advice. We have added the western blot bands for HuR and replaced with the new blot for Akt in **Figure 6C**.

7. *“Fig. 7A. Please show also IHC data from CTR animals under a normal diet.”*

Response: We are very sorry that we don't understand this question clearly. We have added the data including H&E and oil-red staining photos of CTR animals under a normal diet.

We have corrected the following sentence in Page 10.

“HFD-fed CTR mice showed more lipid accumulation in the liver than NCD-fed CTR mice and less lipid accumulation than HuR^{AKO} mice under an HFD diet (**Figure 7A**).”

8. *“Fig. 7D. Please show also IHC data after staining of HuR to test whether macrophages are the primary source of HuR in adipose tissue from HuR knock outs (see also comment # 2).”*

Response: Many thanks for this important question. We examined the expression of HuR in epiWAT, subWAT and BAT by immunohistochemistry assay (**Supplementary Figure 1E**), which indicated that there are a few cells expressing HuR in the adipose tissue from HuR^{AKO} mice. They were proved to mainly be macrophages by F4/80 immunohistochemistry staining (**Figure 7D**). Thus, macrophages are the primary source of HuR in adipose tissue from HuR^{AKO} mice.

9. *“Legend of Figs 7F,G. Does the n number (6) correspond to the total number of patients? In the latter case, a statistic analysis to my opinion is not valid since the number of non-obese controls was only 2 (according to what is*

shown in the W.blot in panel F).”

Response: We are very sorry that we didn't write it clearly. We firstly collected twelve samples, six for non-obese controls and six for obese patients, n=6 for each group. We performed the western blot with more samples and replaced with new **Figure 7F**.

Conceptual comments:

1. *“According to results from HuR ko. mice, the authors conclude that HuR has a protective role against high-fat diet-induced obesity and this statement is also declared in the title of the ms. Besides of the genetic model of obesity (ob/ob mice), most of experiments in which the functional impact of HuR was assessed were performed under conditions of a high fat diet (HFD). Thereby, according to data presented in Fig. 1B and Fig. 5B, HFD has a strong suppressive effect on HuR and ATGL abundance similar to genetic depletion of HuR which seems physiologically relevant to avoid from generation of increased free fatty acids as energy substrate (shown in Fig. 1E). Nevertheless, if HuR exerts an overall protective role in diet-induced obesity, I am wondering why it is downregulated instead of being upregulated? Since most of their conclusions rely on data obtained with HuR knock-out mice under a high fat diet, they should additionally show the effects of HFD in wild-type animals.*

Or in other words: They show pathophysiological effects of a gene knockout under experimental conditions (HFD) which per se induce a strong ablation of the gene.

The authors should comment on this phenomenon in the discussion.”

Response: Thanks for this suggestion. We analyzed the glucose tolerance and insulin resistance in CTR mice fed with normal chow diet (NCD) and high fat diet (HFD) for four months. Compared with NCD-fed CTR mice, HFD-fed CTR mice showed impaired glucose tolerance and insulin resistance (**Supplementary Figure 2A-2B**), and more lipid accumulation in liver (**Figure 7A**) with a decreased adipose HuR expression. For the HuR^{AKO} mice with HuR

completely knockout in adipocytes, HFD-fed HuR^{AKO} mice possessed a more deteriorated phenotype in glucose hemostasis (**Supplementary Figure 2A-2B and Figure 6A-6B**) and ectopic lipid deposition (**Figure 7A**) than HFD-fed CTR mice. Thus, HuR plays an overall protective role in diet-induced obesity. A series of changes in mice phenotypes in diet-induced obesity can be attributed to decreased HuR expression.

We have added the following sentence in Page 9.

“Compared with NCD-fed CTR mice, HFD-fed CTR mice showed impaired glucose tolerance and insulin resistance (**Supplementary Figure 2A-2B**), accompanied with a decreased adipose HuR expression.”

We have corrected the following sentence in Page 10.

“HFD-fed CTR mice showed more lipid accumulation in the liver than NCD-fed CTR mice and less lipid accumulation than HuR^{AKO} mice under an HFD diet (**Figure 7A**).”

2. *“Since the mRNA and protein of HuR are quite stable (due to HuR-triggered autoregulation), an important issue which is not addressed in this ms. is the mechanism which is underlying HFD-induced HuR repression. From the literature, HuR is known to be mainly regulated by means of intracellular localization and modulation of RNA-binding with both of these modes are controlled by diverse posttranslational modifications. The authors should at least discuss possible mechanisms which could explain the HFD-induced repression of HuR e.g. transcription, translation, reduction in mRNA or protein half-life, ubiquitin-dependent proteasomal or caspase-triggered degradation (see also comment # 1).”*

Response: Thanks for this suggestion. HuR function is regulated via changes in its subcellular localization, affinity for RNA, abundance, and cleavage, which in turn influence HuR's ability to affect the fate of target mRNAs¹⁻⁴. In this study, the mRNA levels of HuR in epiWAT, subWAT and BAT were decreased from HFD-fed mice (**Supplementary Figure 1B**), which suggesting that

HFD-induced repression of HuR could be because of its mRNA reduction. High fat diet may inhibit the HuR transcription.

We have added the following sentence in Page 12.

“Moreover, HuR function is regulated via changes in its subcellular localization, affinity for RNA, abundance, and cleavage, which in turn influence its ability to affect the fate of target mRNAs³⁵⁻³⁸. In this study, the mRNA levels of HuR in epiWAT, subWAT and BAT were decreased from HFD-fed mice (**Supplementary Figure 1B**), which suggested that HFD-induced repression of HuR could be attributed to its mRNA reduction. High fat diet may inhibit the HuR transcription.”

Minor:

1. *“P. 8 (line 162). Please replace “modification” by “modulation”. The term modification is generally used for post-translational events.”*

Response: Thanks very much. We have corrected.

2. *“Material and Methods: The authors should mention the source of CMLD-2. To the best of my knowledge this inhibitor is no longer commercially available.”*

Response: We purchased the inhibitor CMLD-2 from Millipore and the Catalogue Number has been added in the Material and Methods. It is still commercially available.

3. *“P12, line 245: “hormone-sensitive” instead of “sensitive”*

Response: Thanks very much. We have corrected.

Reviewer #2

1. *“In principal, the paper is interesting showing that HuR-affects adipose tissue function and metabolic disease. However, it is important to note that*

reduced ATGL expression in adipose tissue rather counteracts insulin resistance and liver steatosis. Thus, the phenotype of adipose-specific HuR-deficiency cannot be explained by reduced lipolysis which should be discussed in the manuscript.”

Response: Thanks for this interesting question. In this study, obesity can be explained by blunted lipolysis in some degree. Since the triglycerides within adipocytes constitute >90% of adipocyte volume and is the major volume component of the adipose mass⁵, suppressed lipolysis could cause retained lipids in adipocytes and thereby expand adipose tissue.

So far, there is no consensus opinion about the relationship between ATGL and insulin sensitivity. Recent studies reported that adipose overexpression of ATGL attenuated diet-induced obesity and improved insulin sensitivity⁶. Adipose-specific knockout of sirtuin 6 (Sirt6) in mice exacerbated insulin resistance and inflammation in HFD-induced obesity, in which ATGL expression and lipolysis were inhibited⁷, which was consistent with our study. Thus, suppressed ATGL levels due to HuR knockout may contribute to reduced lipolysis and the phenotype with obesity in the HuR^{AKO} mice. Besides, hepatic and adipocyte inflammation were exacerbated in the HuR^{AKO} mice, which may contribute to the impaired glucose intolerance and insulin resistance. The rising data have demonstrated that immune cells such as macrophages reside in or infiltrate metabolic organs under obese condition can cause the low-grade inflammation that impairs insulin action thus leading to the development of insulin resistance⁸.

We have added the following sentences in Page 13.

“Since the triglycerides within adipocytes constitute >90% of adipocyte volume and is the major volume component of the adipose mass⁴³, suppressed lipolysis could cause retained lipids in adipocytes and thereby expand adipose tissue. So far, there is no consensus opinion about the

relationship between ATGL and insulin sensitivity⁴⁴. Recent studies reported that adipose overexpression of ATGL attenuated diet-induced obesity and improved insulin sensitivity.” “which was consistent with our study. Thus, suppressed ATGL levels due to HuR knockout may contribute to reduced lipolysis and the phenotype with obesity in the HuR^{AKO} mice.”

2. “Accordingly, authors show that many genes are up- or downregulated in adipose tissue. Which of these genes are predicted targets of HuR and which of them affect adipose tissue function?”

Response: Thanks for this question. We consulted the mRNA 3'-UTR region of 113 downregulated genes in HuR-deficient adipose tissue and found that there were **75** genes with those 3'-UTR region containing adenylate-uridylylate-rich element (AREs) as the predicted targets of HuR, among which **9** genes were reported to affect adipose tissue function (**Supplementary Table 2**).

We have added the following sentence in Page **6**.

“We consulted the mRNA 3'-UTR region of 113 downregulated genes and found that there were 75 genes with those 3'-UTR region containing AREs as the predicted targets of HuR, among which 9 genes were reported to affect adipose tissue function (**Supplementary Table 2**).”

3. “*PLIN* plays a key role in the regulation of lipolysis and triglyceride storage, and is strongly downregulated. Is it a HuR-target?”

Response: This is an interesting question. The differentiated 3T3-L1 were infected with lentivirus expressing GFP or HuR followed by PLIN mRNA stability assay. HuR over-expression did not affect the half-life of PLIN (**Supplementary Figure 2D**), which indicating that PLIN is not a HuR target.

We have added the following sentence in Page **9**.

“However, HuR over-expression did not affect the half-life of PLIN

(**Supplementary Figure 2D**), which indicating that PLIN is not a HuR target.”

Minor comments:

1. “11: *respiration exchange rate/ respiratory exchange rate*”

Response: We are sorry for the incorrect expression and have corrected it.

2. “Fig.3: *Quantify changes in oxygen consumption, respiratory exchange ratio and heat production.*”

Response: Thanks for the advice. We have added the quantitative data of changes in oxygen consumption, respiratory exchange ratio and heat production in **Supplementary Figure 3A-3C**.

3. “124: *lipoapoprotein lipase/lipoprotein lipase*”

Response: Thanks very much. We have corrected.

4. “124: “ *β -oxidative genes such as acetyl-CoA carboxylase (ACC)*”. ACC is involved in fatty acid synthesis.”

Response: We are very sorry for the mistake and have corrected.

5. “Fig.4 *Is differentiation unchanged in SVF lacking HuR and 3T3L1 overexpressing HuR?*”

Response: Thank you for this important question. We separated the SVF from HuR^{AKO} mice and its littermates CTR mice. Besides, 3T3-L1 cells were infected with lentivirus expressing GFP or HuR. These cells were differentiated into adipocyte-like cells with similar efficiency between HuR^{AKO} and CTR groups, or between GFP and HuR groups, which indicating that HuR over-expression or knockout didn't affect the adipose differentiation (**Supplementary Figure 3D-3E**).

We have added the following sentence in Page 5.

“Besides, HuR over-expression or knockout didn’t affect the adipose differentiation (**Supplementary Figure 3D-3E**)”

6. “Fig. 7 is there a correlation between BMI with HuR expression?”

Response: This is a very important question. We another collected 13 samples of obese individuals with different BMI and measured HuR expression by western blot. We discovered the tendency quantitatively with a regression line (**Figure 7G**), which indicating that the expression of HuR was reverse consistent with BMI.

We have added the following sentence in Page 11.

“More importantly, we discovered the tendency quantitatively with a regression line (**Figure 7G**), which indicating that the expression of HuR was reverse consistent with body mass index (BMI).”

7. “279: *Which experiments support the hypothesis that "HuR may be a potential therapeutic target for ameliorating obesity and obesity-related metabolic disorders"*”.

Response: Thanks for this question. We are sorry that we didn’t prove the therapeutic role directly by over-expressing HuR *in vivo*. The adipose tissue specific HuR knockout mice showed the adverse effects compared to its CTR littermates, which to some extent suggests the potential protective effects of HuR against obesity and obesity-related metabolic disorders. Furthermore, we over-expressed HuR in differentiated 3T3-L1 cells and ATGL expression was elevated (**Figure 5D**). Consideration that adipose overexpression of ATGL could attenuate diet-induced obesity and improve insulin sensitivity⁶, we speculate that HuR may be a potential therapeutic target for ameliorating obesity and obesity-related metabolic disorders. However, we need more experiments to further demonstrate this concept.

References

- 1 Grammatikakis, I., Abdelmohsen, K. & Gorospe, M. Posttranslational control of HuR function. *Wiley Interdiscip Rev RNA* **8**, doi:10.1002/wrna.1372 (2017).
- 2 Doller, A. *et al.* Posttranslational modification of the AU-rich element binding protein HuR by protein kinase Cdelta elicits angiotensin II-induced stabilization and nuclear export of cyclooxygenase 2 mRNA. *Molecular and cellular biology* **28**, 2608-2625, doi:10.1128/MCB.01530-07 (2008).
- 3 Yi, J. *et al.* Reduced nuclear export of HuR mRNA by HuR is linked to the loss of HuR in replicative senescence. *Nucleic Acids Res* **38**, 1547-1558, doi:10.1093/nar/gkp1114 (2010).
- 4 Abdelmohsen, K., Kuwano, Y., Kim, H. H. & Gorospe, M. Posttranscriptional gene regulation by RNA-binding proteins during oxidative stress: implications for cellular senescence. *Biol Chem* **389**, 243-255, doi:10.1515/BC.2008.022 (2008).
- 5 Arner, P. & Langin, D. Lipolysis in lipid turnover, cancer cachexia, and obesity-induced insulin resistance. *Trends Endocrinol Metab* **25**, 255-262, doi:10.1016/j.tem.2014.03.002 (2014).
- 6 Ahmadian, M. *et al.* Adipose overexpression of desnutrin promotes fatty acid use and attenuates diet-induced obesity. *Diabetes* **58**, 855-866, doi:10.2337/db08-1644 (2009).
- 7 Kuang, J. *et al.* Fat-Specific Sirt6 Ablation Sensitizes Mice to High-Fat Diet-Induced Obesity and Insulin Resistance by Inhibiting Lipolysis. *Diabetes* **66**, 1159-1171, doi:10.2337/db16-1225 (2017).
- 8 Lauterbach, M. A. & Wunderlich, F. T. Macrophage function in obesity-induced inflammation and insulin resistance. *Pflugers Arch* **469**, 385-396, doi:10.1007/s00424-017-1955-5 (2017).

Reviewers' comments:

Reviewer #1 (Remarks to the Author):

Most of the points I raised have been addressed. However, some critical points were not adequately addressed. Unless, these points are not changed satisfactory, I cannot recommend acceptance of the paper.

1. (concerns # 6). In contrast to their statement in the response letter, the blot for total Akt in Figure 6C (middle panel) was not replaced and seems not representative for the graph shown at the bottom!

The reduction in p-Akt levels in liver goes parallel with a decrease in total Akt levels. Moreover, I am still not happy with the statement on p.9 "Insulin-induced Akt phosphorylation was attenuated in adipose tissue, liver and skeletal muscle of HuR AKO mice" I think this statement is misinterpreted. In contrast to their statement, the stimulus mediated-induction in Akt phosphorylation seems even increased (compare black bars in HuR AKO in – vs. + in liver and muscle tissue) mainly because of the total loss in basal Akt-phosphorylation in both nonadipose tissues. To my opinion, this clearly demonstrates that in these tissues HuR depletion leads to a reduction of basal Akt signaling under a HFD, but to an increased sensitivity to insulin. The diversity in different tissues should be mentioned. It is also inept that the application of samples in the three gels differs (left gel is loaded "- - + +" whereas the other gels are loaded in an alternate manner "- + - +"). This will definitely increase the risk of misinterpretations mainly if considering that the final size of figure captions will be very small in the printed version.

2. (#4). Fig. 5F. The inhibitory effects by CMLD-2 on HuR mRNA binding would become more prominent and more clear if less cycles are used for amplification of cDNA.

3. (# 7D). In contrast to their statement in the response letter, I cannot see a positive staining for F4/80 in HuRAkomice. In order to prove a real double staining of both proteins within the same cell, the authors should use fluorescently labelled antibodies by confocal microscopy.

4. the newly added sentence on p.12 should be changed to "High fat diet may either interfere with the transcription of HuR or, alternatively, increase the decay of HuR mRNA. This important issue needs to be addressed by future studies."

Reviewer #2 (Remarks to the Author):

Authors have carefully considered reviewer's comments and improved their manuscript. I have only minor additional corrections/questions?

18 thereby control obesity and metabolic syndrome(s)

35 patatin-like phospholipase ...patatin-like

181 Moreover, (in) consistent with the...

Figure 7G: Is this protein or mRNA expression? Is it normalized to tissue protein/weight or a house keeping gene?

Responds to the reviewer's comments:

Reviewer #1:

1. *“(concerns # 6). In contrast to their statement in the response letter, the blot for total Akt in Figure 6C (middle panel) was not replaced and seems not representative for the graph shown at the bottom! The reduction in p-Akt levels in liver goes parallel with a decrease in total Akt levels. Moreover, I am still not happy with the statement on p.9 “Insulin-induced Akt phosphorylation was attenuated in adipose tissue, liver and skeletal muscle of HuR AKO mice” I think this statement is misinterpreted. In contrast to their statement, the stimulus mediated-induction in Akt phosphorylation seems even increased (compare black bars in HuR AKO in – vs. + in liver and muscle tissue) mainly because of the total loss in basal Akt-phosphorylation in both non-adipose tissues. To my opinion, this clearly demonstrates that in these tissues HuR depletion leads to a reduction of basal Akt signaling under a HFD, but to an increased sensitivity to insulin. The diversity in different tissues should be mentioned.*

It is also inept that the application of samples in the three gels differs (left gel is loaded “- - + +” whereas the other gels are loaded in an alternate manner “- + - +”). This will definitely increase the risk of misinterpretations mainly if considering that the final size of figure captions will be very small in the printed version.”

Response: Thanks for this question. Firstly, we are very sorry for that we forgot to replace the total Akt blot in **Figure 6C (middle panel)**. And we have replaced it with a representative blot.

Besides, for the statement on Page 9, we seriously considered your opinion and consulted literatures. Indeed, although the basal p-Akt level is lower in the liver and muscle of HuR^{AKO} mice, the increase fold of p-Akt under insulin-stimulated condition is more than that of CTR mice. Therefore, we have reason to think that in liver and muscle, HuR depletion leads to an increased

sensitivity to insulin. In adipose tissues, there is no significant difference in basal Akt level between these two groups. However, the p-Akt level to insulin is lower in HuR^{AKO} mice compared with CTR mice. Above all, we think that the glucose intolerance and insulin resistance in HuR^{AKO} mice can be partly attributed to suppressed p-Akt signaling to insulin in adipose tissue. As for the increased sensitivity to insulin in the liver and muscle of HuR^{AKO} mice, we need to explore the specific mechanism in future studies.

Furthermore, to avoid the risk of misinterpretations caused by different loading orders, we repeated the western blot of adipose tissue samples and have replaced with new blots in **Figure 6C (left panel)**.

We have replaced with the following sentences in Page 9 and 14.

“Insulin-induced Akt phosphorylation was attenuated in adipose tissue of HuR^{AKO} mice, indicating that HuR ablation in adipocytes could exacerbate HFD-induced insulin resistance in adipose tissue. However, given a significant reduction of basal Akt signaling and an increased fold under insulin-induced situation in liver and muscle, adipose-specific HuR depletion led to their increased sensitivity to insulin under HFD (**Figure 6C-6D**).”

“In our study, HuR deletion deteriorated insulin resistance and inflammation infiltration in adipose tissue.”

2. “(#4). *Fig. 5F. The inhibitory effects by CMLD-2 on HuR mRNA binding would become more prominent and more clear if less cycles are used for amplification of cDNA.*”

Response: Thanks for this suggestion. We performed the RIP experiment with decreased cycle numbers in amplification of cDNA and have replaced with a new **Figure 5F**, which showed the inhibitory effect of CMLD-2 more clearly.

3. “(# 7D). *In contrast to their statement in the response letter, I cannot see a positive staining for F4/80 in HuR^{AKO} mice. In order to prove a real double*

staining of both proteins within the same cell, the authors should use fluorescently labelled antibodies by confocal microscopy. ”

Response: Thanks for this suggestion. We performed the double-labelling immunofluorescence assay and detected the expression of HuR and F4/80 by fluorescence microscopy. We found that F4/80-positive macrophages are the primary source of HuR in adipose tissue from HuR^{AKO} mice. And we have replaced with a new **Figure 7D**.

We have replaced with the following sentences in Page 11.

“Immunofluorescent staining of HuR and macrophage marker F4/80 suggested that macrophages are the primary source of HuR in adipose tissue from HuR^{AKO} mice and revealed increased crown-like structures in epididymal adipose tissue of HFD-fed HuR^{AKO} mice (**Figure 7D**).”

4. *“the newly added sentence on p.12 should be changed to “High fat diet may either interfere with the transcription of HuR or, alternatively, increase the decay of HuR mRNA. This important issue needs to be addressed by future studies.”*

Response: Thanks for this important suggestion. We have replaced with the more rigorous explanation in Page 13.

Reviewer #2

1. *“18 thereby control obesity and metabolic syndrome(s)”*

Response: Thanks very much. We have corrected.

2. *“35 patalin-like phospholipase ...patatin-like”*

Response: We are very sorry for the mistake and have corrected.

3. *“181 Moreover, (in) consistent with the...”*

Response: Thanks very much. We have corrected.

4. *“Figure 7G: Is this protein or mRNA expression? Is it normalized to tissue protein/weight or a house keeping gene?”*

Response: Thanks for the question. We are sorry that we didn't make it clear. The Figure 7G showed the correlation between the normalized HuR protein expression and BMI. We examined the expression of HuR and housing-keeping protein β -actin by western blot assay, and then normalized the HuR protein levels to β -actin expression. Finally we drew the scatter diagram and regression line based on the normalized HuR protein expression and BMI data.

We have replaced with the following sentence in Page 11.

“More importantly, we discovered the tendency quantitatively with a regression line (Figure 7G), which indicated that the normalized protein expression of HuR was reversely correlated with body mass index (BMI).”

REVIEWERS' COMMENTS:

Reviewer #1 (Remarks to the Author):

All of points raised have been addressed adequately. Therefore, I have no further comments and questions.

Congratulations to this exiting set of new data!

Reviewer #2 (Remarks to the Author):

Authors have revised their study according to the reviewer's suggestions and substantially improved their manuscript. I have no further comments

Responds to the referees' comments

REVIEWERS' COMMENTS:

Reviewer #1 (Remarks to the Author):

1. All of points raised have been addressed adequately. Therefore, I have no further comments and questions.

Congratulations to this exiting set of new data!

Response: Thanks for your previous great suggestions. It helps us a lot to improve our work.

Reviewer #2 (Remarks to the Author):

1. Authors have revised their study according to the reviewer's suggestions and substantially improved their manuscript. I have no further comments.

Response: Thank you for your very helpful suggestions. It definitely improved our manuscript.